# FlexTrain: A Dynamic Training Framework for Heterogeneous Devices Environments

**Mert Unsal**
Huawei
mert.unsal1@huawei.com

**Ali Maatouk**
Huawei
ali.maatouk@huawei.com

**Antonio De Domenico**
Huawei
antonio.de.domenico@huawei.com

**Nicola Piovesan**
Huawei
nicola.piovesan@huawei.com

**Fadhel Ayed**
Huawei
fadhel.ayed@huawei.com

## Abstract

As deep learning models become increasingly large, they pose significant challenges in heterogeneous devices environments. The size of deep learning models makes it difficult to deploy them on low-power or resource-constrained devices, leading to long inference times and high energy consumption. To address these challenges, we propose FlexTrain, a framework that accommodates the diverse storage and computational resources available on different devices during the training phase. FlexTrain enables efficient deployment of deep learning models, while respecting device constraints, minimizing communication costs, and ensuring seamless integration with diverse devices. We demonstrate the effectiveness of FlexTrain on the CIFAR-100 dataset, where a single global model trained with FlexTrain can be easily deployed on heterogeneous devices, saving training time and energy consumption. We also extend FlexTrain to the federated learning setting, showing that our approach outperforms standard federated learning benchmarks on both CIFAR-10 and CIFAR-100 datasets.

## 1 Introduction

Deep learning has emerged as the leading paradigm in machine learning, thanks to its outstanding performance in various tasks, including computer vision, natural language processing, and speech recognition, among others [LeCun et al., 2015]. Its success is primarily attributed to the ability to learn high-level features and representations from raw data, by virtue of using deep neural networks. In fact, deep learning models have demonstrated state-of-the-art results in several benchmark datasets and have achieved a level of accuracy that was previously unattainable with traditional machine learning approaches [Krizhevsky et al., 2017][He et al., 2016]. With the increasing size of datasets, the size of deep learning models has also been increasing, often reaching billions of parameters, to capture more complex and subtle patterns in the data. The trend towards larger models has been evident in several recent studies, including the development of models such as GPT-3 and GPT-4, which contains billion parameters [Brown et al., 2020][OpenAI, 2023]. These models have achieved state-of-the-art performance in a wide range of NLP tasks, demonstrating the power of large ML models in solving complex real-world problems.

Workshop on Advancing Neural Network Training (WANT) at NeurIPS 2023

However, as machine learning models become increasingly large, they pose significant challenges, particularly in heterogeneous devices environments. The size of deep learning models, which can often range from tens of millions to billions of parameters, makes it difficult to deploy these models on low-power or resource-constrained devices such as smartphones, tablets, and edge devices. These devices may not have the necessary computational power, storage, or memory to run these models efficiently, leading to long inference times and high energy consumption. Additionally, these challenges are amplified in federated settings where the training process is decentralized across multiple devices, with each device contributing its local data to the training process while maintaining data privacy [Kairouz et al., 2021]. In these settings, the large size of deep learning models can result in high communication overhead between devices and the central server, leading to slow convergence rates, and suboptimal models [McMahan et al., 2017], thus prompting works in the so-called communication-aware machine learning frameworks (e.g., [Ayed et al., 2023]). Furthermore, some devices may have limited computational resources and may not be able to participate in the training process, leading to bias in the final model due to a lack of representative data [Yang et al., 2019].

Several methods have been proposed to address these challenges, including model compression, quantization, and sparsification. Model compression techniques focus on reducing the size of the model without compromising its performance. One popular approach is knowledge distillation, which involves training a smaller model to emulate the output of a larger model [Hinton et al., 2015]. Another approach is quantization, which reduces the precision of the model weights to reduce its storage and memory requirements (e.g., integer-only arithmetics as in [Jacob et al., 2018]). Sparsification techniques, on the other hand, aim to prune the model's connections or weights, leading to a smaller and more efficient model [Guo et al., 2016]. In this paper, we take a different approach to addressing the challenges posed by large models in heterogeneous devices environments. Specifically, we introduce FlexTrain, a framework that is designed to accommodate the diverse storage and computational resources available on different devices during the training phase. The deployment of deep learning models is thus made more efficient by our framework, which accounts for device constraints, reduces communication costs in time-sensitive contexts, and ensures integration with a range of devices. Furthermore, FlexTrain can be used in conjunction with other techniques, such as quantization and compression, to achieve further improvements in flexibility and efficiency. Concretely, our contributions are twofold:

- We propose two key techniques, active layers sampling and auto-distillation, which form the core of FlexTrain for residual neural architectures such as ResNets and Transformers [He et al., 2016][Vaswani et al., 2017]. Active layers sampling enables us to dynamically select a subset of residual layers during training, allowing us to train models that are better suited for deployment on heterogeneous devices. auto-distillation, on the other hand, enables us to distill knowledge learned by larger models into smaller ones, resulting in better performance on low-capacity devices. We demonstrate the effectiveness of FlexTrain on the CIFAR-100 dataset, where we show that by sacrificing only a small percentage of accuracy, a single global model trained with FlexTrain can be easily deployed on heterogeneous devices, saving training time and energy consumption.

- We extend FlexTrain to the federated learning setting and show that our approach outperforms standard federated learning benchmarks on both CIFAR-10 and CIFAR-100 datasets. Federated FlexTrain is able to achieve higher accuracy by effectively sharing knowledge learned from the weaker devices to the more capable ones. In addition, we discuss how this advantage is further amplified in scenarios involving a large number of devices and non-independent and identically distributed (non-i.i.d.) data distributions, highlighting the practical implications of FlexTrain for real-world applications of federated learning.

## 2 FlexTrain Methodology

In this section, we begin by describing the layer-wise sampling and activation techniques used in FlexTrain, which enable us to achieve the desired flexibility. We then present our proposed auto-distillation method, which facilitates the exchange of knowledge between features learned by the various sampled models, resulting in even better overall performance. With these techniques, we can create a highly optimized model that can be deployed efficiently while minimizing communication costs in time-sensitive environments and ensuring seamless integration with diverse devices.

## 2.1 Active Layers Sampling

Consider a dataset $\mathcal{D} = \mathcal{X} \times \mathcal{Y} \subseteq \mathbb{R}^{d_x} \times \mathbb{R}^{d_y}$ consisting of $n$ (input, label) pairs $\{(x_i, y_i)\}_{1 \leq i \leq n}$ and a neural network model $\mathcal{N}$ of a given architecture belonging to the family

$$\mathcal{N}^\theta = \{y_{out}(\cdot; W) : \mathcal{X} \to \mathcal{Y} \mid W \in \mathcal{W}^\theta\} \tag{1}$$

parameterized by the *parameter space* $\mathcal{W}^\theta$, where $\theta \in \Theta$ is a fixed *hyper-parameter* that represents architecture-related quantities such as the depth of the network $K$. Let $\ell : \mathbb{R}^{d_x} \times \mathbb{R}^{d_y} \to \mathbb{R}$ be a loss function, e.g. quadratic loss, cross-entropy loss etc., and let us define the model loss for a single sample $(x, y) \in \mathcal{D}$ as

$$\mathcal{L}(x, y; W) = \ell(y_{out}(x; W), y), \tag{2}$$

where $W = (W_k)_{1 \leq k \leq K}$ and $W_k$ are the adjustable parameters of layer $k$. Given the above loss function, the aim of the learning procedure is to find the parameters $W$ that minimize the empirical risk

$$\min_{W \in \mathcal{W}^\theta} \mathcal{L}(W) \triangleq \frac{1}{n} \sum_{i=1}^{n} \ell\big(y_{out}(x_i; W), y_i\big). \tag{3}$$

The minimization problem stated in equation (3) is commonly tackled through numerical techniques, with gradient-based methods such as Stochastic Gradient Descent [Robbins and Monro, 1951] and Adam [Kingma and Ba, 2014] being common choices. To achieve low generalization error, a deep network with a substantial number of parameters in the weight matrix $W$ is necessary, as predicted by the power-law scaling of neural networks. For instance, state-of-the-art transformer-based language models like LLaMA and PaLM comprise up to 65B and 540B parameters, respectively, and consist of 80 and 118 layers [Touvron et al., 2023][Chowdhery et al., 2022]. This trend of increasingly large machine learning models is also evident in other tasks such as vision [Villalobos et al., 2022]. For instance, ResNet-110, a popular architecture for image classification, comprises 1.7M parameters [He et al., 2016]. The large size of these models presents a significant obstacle to deployment on edge devices, particularly in time-sensitive environments where several models may be needed. As a result, it is essential to devise efficient techniques for deploying these models in such environments.

One potential solution to address this challenge is to train models of varying sizes independently and then deploy them based on the edge device's capabilities. For example, LLaMA models offer multiple size options ranging from 7B to 65B parameters, allowing for more flexible deployment options [Touvron et al., 2023]. Nonetheless, training multiple large models can quickly become impractical due to the associated cost and environmental footprint. FlexTrain is a solution that tackles this issue by sampling various configurations of the same neural network $\mathcal{N}$ during the training process to create a flexible model with minimal accuracy loss compared to the full model. These configurations are obtained by selectively deactivating a certain number of layers starting from the deepest layers. The deep-to-shallow deactivation approach is motivated by the desire to allow the basic features to be learned during the training process and to enable deeper layers to build upon them and learn more complex features. This makes our approach particularly well-suited for residual neural architectures such as Residual Networks (ResNets) and Transformers, which are widely used in modern deep-learning applications. As a result, we will concentrate our attention on ResNets and Transformers in the rest of the paper. Figure 1 provides a visual illustration of this concept, with blue and white blocks representing active and inactive blocks, respectively.

With this in mind, we let $\mathcal{N}_k$ denote the neural network comprising the first $k$ layers with their corresponding weights $W_k$ activated, while the rest of the blocks function as identity. We use $\tilde{W}_k$ to denote the weights of the first $k$ layers, including the pre-processing layer and the decision layer (e.g., classification layer). Next, we let $\boldsymbol{\pi} = [\pi_1, \ldots, \pi_K]$ represent the activation distribution of the network's layers over the set $\{1, \ldots, K\}$. Specifically, $\pi_K$ denotes the probability that all layers are activated during any training epoch, while $\pi_1$ denotes the probability that only the first layer is activated. The choice of this specific distribution is intricately tied to the constraints of edge devices. This is because when a particular model configuration is more commonly selected, it can lead to better performance on devices that can only handle that configuration. By carefully selecting the distribution that best suits the limitations of the edge devices, we can ensure optimal performance and efficiency for the system as a whole. For example, one possible option for $\pi_K$ is to set it as the ratio of devices that can afford deployment of the full model. Given this, one can conclude that FlexTrain

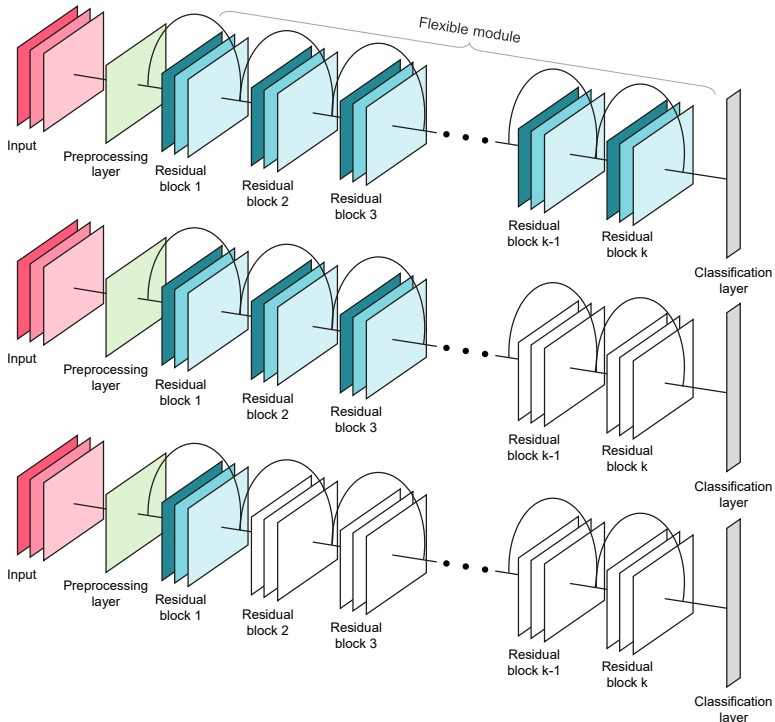

Figure 1: Illustration of the active layers sampling procedure used in FlexTrain for three different configurations. The blue blocks represent activated layers, while the white blocks represent inactive layers that operate as the identity function.

aims to minimize the following loss function

$$\overline{\mathcal{L}}(W) = \mathbb{E}_{k \sim \boldsymbol{\pi}}[\mathcal{L}(\tilde{W}_k)] \triangleq \sum_{k=1}^{K} \pi_k \mathcal{L}(\tilde{W}_k). \tag{4}$$

This approach offers two major benefits:

- **Reduced Training Time:** The reduction in training time is due to the shorter forward and backward propagation, which means that the training time no longer scales with the full depth of the network. Instead, it scales with the shorter expected depth of the network. This can be seen by defining the expected adjusted parameters at any training epoch as $\overline{r} = \frac{\sum_{k=1}^{K} \pi_k A_k}{A}$, where $A_k$ denotes the number of parameters making up $\tilde{W}_k$, and $A$ is the total number of tunable parameters of the neural network $\mathcal{N}$. Given that $\pi_K < 1$ whenever at least a device cannot afford to run the full model, one can conclude that $\overline{r} < 1$, showcasing that the training time is indeed reduced.

- **Flexible Deployment:** During the training process, different model configurations are encountered, which makes deployment simple. If a device has limited storage and processing capabilities and can only accommodate a model of size $A^*$, the first $k^*$ layers of the trained model that ensure that this constraint is met are deployed. If the device's capabilities improve with time, additional layers can be provided to the device without the need for any updates to the previously delivered $k^*$ layers. Another perspective on this flexibility is that the device can benefit from using a small model while awaiting the delivery of the larger model, which is highly advantageous in time-sensitive environments. All this demonstrates the flexibility of the trained model and showcases the adaptability of the approach to different deployment scenarios.

These benefits are crucial, but it's also essential to ensure that the accuracy of any model configuration matches that obtained when training the corresponding model size from scratch. This leads us to the next section.

**Remark.** We chose to focus on neural networks because of the widespread use of models such as ResNets and Transformers in modern applications involving vision and text tasks. Nonetheless, it is worth noting that FlexTrain can also be applied to other models that use a boosting approach that shares similarities with residual connections, allowing them to gain similar flexibility advantages.

## 2.2 Auto-distillation

The flexibility provided by FlexTrain can only be of interest if no significant loss in accuracy is incurred compared to the case where models of varying sizes are independently trained on the data. To further push toward achieving this goal, and given the dynamic nature of the active layers sampling during the training process, we use a modification of the knowledge distillation technique that is commonly used to transfer knowledge from a teacher model to a student model as described in the literature [Hinton et al., 2015] [Zhang et al., 2019]. Particularly, at each training epoch and for each training sample $(x, y) \in \mathcal{D}$, we add a term to the loss function reported in eq. (2) representing the squared $\mathcal{L}_2$ distance between the final activation layer of the current configuration $f_{\tilde{W}_k}(\cdot)$ and that of a larger configuration $f_{\tilde{W}_{k'}}(\cdot)$ as follows

$$\ell_{\text{dist}}(x, y; \tilde{W}_k) = \ell(x, y; \tilde{W}_k) + \beta \|f_{\tilde{W}_k}(x) - f_{\tilde{W}_{k'}}(x)\|_2^2, \quad k \le k' \le K, \tag{5}$$

where $\beta$ is a fixed constant. Note that the term $f_{\tilde{W}_{k'}}(x)$ is not differentiated when propagating the loss gradient across $\mathcal{N}_k$. Ideally, one would distill knowledge from the largest configuration $\mathcal{N}$, which contains the most knowledge, to the smaller configurations. However, this can be computationally expensive for large neural networks. Our experiments in Section 4 show that fixing $k'$ to $k+1$ already provides a good performance benchmark. Given all of this, we summarize in Algorithm 1 the totality of the FlexTrain framework.

---

**Algorithm 1:** FlexTrain

**Input:** Model $\mathcal{N}$, distribution $\boldsymbol{\pi}$, learning rate $\alpha$, dataset $\mathcal{D}$

1   Initialize $W$;
2   **while** *not converged* **do**
3      Sample configuration $k \sim \boldsymbol{\pi}$;
4      Sample batch $(\mathbf{x}, \mathbf{y}) \sim \mathcal{D}$ of size $B$;
5      $\tilde{W}_k \leftarrow \tilde{W}_k - \alpha \frac{1}{B} \sum_{j=1}^B \nabla_{\tilde{W}_k} \ell_{\text{dist}}(x, y; \tilde{W}_k)$;
6   **end while**

---

## 3   Federated FlexTrain

The heterogeneity of devices in federated settings poses a significant challenge in large-scale deep-learning applications [Kairouz et al., 2021]. This is compounded by the fact that data cannot be shared, making it less desirable to train multiple-sized models simultaneously as data from less-capable devices will only contribute to training smaller models, resulting in skewed data and limited richness for larger models, especially in non-i.i.d data cases. To address this challenge, a flexible training algorithm that can adapt to different training and inference scenarios is essential to leverage the data of all devices. In recognition of the importance of federated settings in a broad range of applications such as mobile keyboard prediction [Hard et al., 2018], we extend our FlexTrain method to the federated settings to enable the development of adaptable models that can improve overall performance.

In the realm of federated learning, the foremost objective is to minimize a loss function across multiple devices while maintaining the privacy of the data. Typically, this involves defining the loss function as the sum of the individual loss functions of all devices, weighted by the number of data samples on each device. Specifically, given a dataset $\mathcal{D}_j$ on device $j$ among $J$ devices, a neural network model $\mathcal{N} \in \mathcal{N}^\theta$, and a loss function $\mathcal{L}(\cdot, \cdot; W)$ as described in eq. (2), the overall loss function in federated settings is given by

$$\mathcal{L}^{\text{Fed}}(W) = \sum_{j=1}^J \frac{|\mathcal{D}_j|}{|\mathcal{D}|} \mathcal{L}_j^{\text{Fed}}(W), \tag{6}$$

where $\mathcal{L}_j^{\text{Fed}}(W) \triangleq \frac{1}{|\mathcal{D}_j|} \sum_{(x,y) \in \mathcal{D}_j} \mathcal{L}(x, y; W)$. Due to device heterogeneity, the central idea behind Federated FlexTrain is to provide each device with a model configuration (i.e., a fraction of the full model) that matches its resource capabilities, enabling it to perform local training epochs on its own dataset. The updated weights of the model configuration are then transmitted to the central server where they are appended to the remaining weights of the full model, and aggregated with the updates of other devices akin to the FedAvg algorithm [McMahan et al., 2016]. This approach enables the implementation of the centralized FlexTrain algorithm in a distributed manner, taking into account the resource constraints of all devices during training and inference. Following suit the notations depicted in Section 2, and by letting $\boldsymbol{\pi} = [\pi_1, \ldots, \pi_K]$ denote a probability distribution where $\pi_k$ represents the ratio of devices that can afford deployment of the first $k$ layers of the neural network $\mathcal{N}$, we can conclude that Federated FlexTrain minimizes the expected loss over the distribution of different model configurations afforded by the devices detailed below

$$\overline{\mathcal{L}}^{\text{Fed}}(W) = \sum_{j=1}^{J} \frac{|\mathcal{D}_j|}{|\mathcal{D}|} \mathbb{E}_{k_j \sim \boldsymbol{\pi}}[\mathcal{L}_j^{\text{Fed}}(\tilde{W}_{k_j})]. \tag{7}$$

**Auto-distillation.** Integrating the distillation concept introduced in Section 2.2 into the federated case is desirable, but challenging. Devices running different configurations cannot access a shared dataset, making it difficult to apply the same distillation techniques. To address this issue, we propose a workaround, where each device (except for the weakest ones) distills the previous configuration using their local dataset. We achieve this by adding a term to the loss function reported in eq. (2) representing the squared $\mathcal{L}_2$ distance between the final activation layer of the current configuration $f_{\tilde{W}_k}(\cdot)$ affordable by the device and that of a one-step smaller configuration $f_{\tilde{W}_{k-1}}(\cdot)$ as follows

$$\ell_{\text{dist}}^{\text{Fed}}(x, y; \tilde{W}_k) = \ell(x, y; \tilde{W}_k) + \beta \| f_{\tilde{W}_k}(x) - f_{\tilde{W}_{k-1}}(x) \|_2^2, \tag{8}$$

where $\beta$ is a fixed constant. Note that the term $f_{\tilde{W}_k}(x)$ is not differentiated when propagating the loss gradient across $\mathcal{N}_k$. By adding this term, knowledge is distilled from the larger configuration to the smaller one, enabling smaller models to leverage the advanced capabilities of more powerful devices. Given all of this, we summarize in Algorithm 2 the totality of the FedFlexTrain framework.

---

**Algorithm 2:** Federated FlexTrain

**Input:** Neural Network $\mathcal{N}$, learning rate $\alpha$, local datasets $\mathcal{D}_j$, local epochs $l$

1 **Server:**
2 Initialize $W$;
3 **while** *not converged* **do**
4      Sample devices $C \subseteq \{1, \ldots, J\}$;
5      Ask devices $j \in C$ their model configuration $k_j$ and send $\tilde{W}_{k_j}$;
6      Receive updated $\tilde{W}'_{k_j}$;
7      Aggregate models $W \leftarrow \frac{1}{|C|} \sum_{j \in C} W'_{k_j}$
8 **end while**

9 **Device $j$:**
10 Request the main server for model configuration $k_j$;
11 Receive $\tilde{W}_{k_j}$ from the main server;
12 $counter \leftarrow 0$;
13 **while** *counter $< l$* **do**
14      **foreach** *batch $(x, y) \sim \mathcal{D}_j$ of size $B$* **do**
15          $\tilde{W}_{k_j} \leftarrow \tilde{W}_{k_j} - \alpha \frac{1}{B} \sum_{b=1}^{B} \nabla_{\tilde{W}_{k_j}} \ell_{\text{dist}}^{\text{Fed}}(x^{(b)}, y^{(b)}; \tilde{W}_{k_j})$
16      **end foreach**
17      $counter \leftarrow counter + 1$;
18 **end while**
19 Send updated $\tilde{W}_{k_j}$ to the main server

---

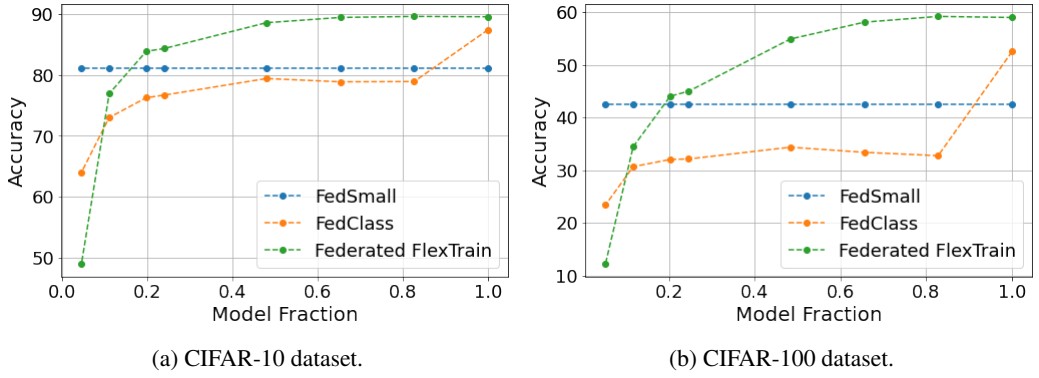

(a) CIFAR-10 dataset.

(b) CIFAR-100 dataset.

Figure 2: Test accuracy comparison of Federated FlexTrain and several benchmark algorithms on (a) CIFAR-10 dataset and (b) CIFAR-100 dataset.

# 4 Experimental Results

We conducted two sets of experiments to demonstrate the efficiency, flexibility, and model capability of FlexTrain. Firstly, we compared the accuracy performance of centralized FlexTrain on ResNet and Visual Transformers architectures [He et al., 2016][Dosovitskiy et al., 2020] for the CIFAR-100 dataset with a benchmark where models of varying sizes were independently trained on the data. Second, we evaluated the accuracy performance of FlexTrain in federated settings using ResNet architectures and compared it to several federated learning benchmarks.

**Centralized Settings.** We applied our proposed FlexTrain framework to train a ResNet-56 model on the CIFAR-100 dataset, setting the number of epochs to 160. To sample the active layers, we employed the following strategy: 1) training the full model with probability 0.5, 2) updating only the first 35% of the parameters with probability 0.25, and 3) updating only the first 15% of the parameters with probability 0.25. We selected the propor-

Table 1: Comparison of test accuracy (%) for ResNet models on CIFAR-100 dataset.

| Model Size Equivalence | Independent | FlexTrain | Single |
|---|---|---|---|
| ResNet-56 | $69.9 \pm 0.85$ | $68.6 \pm 0.64$ | $69.9 \pm 0.85$ |
| ResNet-20 | $68.3 \pm 0.18$ | $63.3 \pm 0.86$ | $6.5 \pm 0.66$ |
| ResNet-8 | $58.5 \pm 1$ | $54.9 \pm 0.66$ | $2.6 \pm 0.53$ |

tions of active model size to be roughly equivalent to three different benchmarks, namely ResNet-56, ResNet-20, and ResNet-8, which were trained independently on the whole dataset. This allowed for a fair comparison between FlexTrain and the multiple independent models approach. In addition, we also conducted another analysis of a ResNet-56 model trained on the CIFAR-100 dataset. Specifically, we evaluated the accuracy performance of different portions of the model, with the portions chosen to be equivalent in size to ResNet-20 and ResNet-8. This allows us to gain insights into the performance of the single model at different levels of abstraction. Our results, as shown in Table 1, indicate that a single model approach is not suitable for deployment in heterogeneous device environments. In contrast, our FlexTrain approach outperformed the single model approach for smaller models, while delivering comparable results for the full model. These results underscore the importance of incorporating flexible training methods that account for the heterogeneity of the environment during the training process. Furthermore, FlexTrain demonstrated a reduced computational requirement, utilizing only 62% of the FLOPs compared to the single model approach and highlighting the diminished complexity of the training procedure employed by FlexTrain. We also compared our approach to the independent models approach and found that while our approach achieved slightly lower accuracy, it only required one training procedure, compared to three separate procedures for the independent models approach. Additionally, our approach required only 41% of the FLOPs needed by the independent models approach, resulting in significant savings in training time and energy. This trade-off between a slight decrease in accuracy and significant savings in resources is particularly advantageous for large-scale deep-learning applications.

Similar to the ResNet architecture experiments, we applied our proposed FlexTrain framework to train a ViT-3 model on the CIFAR-100 dataset, setting the number of epochs to 300. To sample the active layers, we implemented the following strategy: 1) with a probability of 0.5, we trained the full model, 2)

Table 2: Comparison of test accuracy (%) for visual transformers models on CIFAR-100 dataset.

| Model Size Equivalence | Independent | FlexTrain | Single |
|---|---|---|---|
| ViT-1 | $39.3 \pm 0.43$ | $35.9 \pm 0.6$ | $5.8 \pm 0.3$ |
| ViT-2 | $49.7 \pm 0.12$ | $47.5 \pm 0.2$ | $27.5 \pm 0.83$ |
| ViT-3 | $52.8 \pm 0.85$ | $50.5 \pm 0.26$ | $52.8 \pm 0.85$ |

with a probability of 0.25, we updated only the first 2 blocks of the transformer, and 3) with a probability of 0.25, we updated only the first block. Similarly, this activation procedure allows us to compare FlexTrain with the multiple independent models approach. Our results, presented in Table 2, led to conclusions similar to those drawn for the case of ResNet experiments. In this case as well, we can conclude that deploying a single model approach in heterogeneous device environments is ill-advised. Furthermore, our approach, compared to the independent models approach, achieves slightly lower accuracy but requires only one training procedure instead of three separate procedures. Importantly, our approach demanded only 37% of the FLOPs needed by the independent models approach, resulting in significant savings in training time and energy. Thus, these conclusions extend beyond the ResNet architecture, demonstrating their broader applicability.

**Federated Settings.** We conducted experiments using FlexTrain in a federated setting with $J = 20$ devices training a ResNet-56 model on the CIFAR-10 and CIFAR-100 dataset. We set the number of communication rounds to 160 and for each communication round, we set the number of local epochs $l$ to 10. The datasets were split equally among all devices. The capabilities of the devices used in our experiments are listed in Table 3, where $r_j$ represents the proportion of the ResNet-56 size that the device can accommodate, along with the corresponding number of layers it can support starting from the first layer. To evaluate the performance of FlexTrain, we used two distinct benchmarks:

Table 3: Device capabilities in terms of model size and number of layer.

| $r_j$ | $k_j$ | Number of devices |
|---|---|---|
| 5.2% | 8 | 2 |
| 11.7% | 12 | 2 |
| 20.3% | 16 | 2 |
| 24.6% | 18 | 2 |
| 48.3% | 21 | 2 |
| 65.5% | 23 | 2 |
| 82.7% | 25 | 2 |
| 100.0% | 27 | 6 |

1. *FedSmall*: In this approach, the FedAvg algorithm is used to train the largest model that the least capable device can handle.

2. *FedClass*: Another approach is to group devices into different resource classes and train multiple models of different sizes using FedAvg.

The results of our experiments, reported in Table 4, reveal that the Federated FlexTrain method surpasses other benchmarks for both CIFAR-10 and CIFAR-100 datasets in terms of mean device accuracy. To gain deeper insight into these findings, we provide Fig. 2, which displays the mean accuracy of devices as a func-

Table 4: Mean device test accuracy (%) for the considered federated benchmarks on CIFAR-10 and CIFAR-100 datasets.

| Datasets | Federated FlexTrain | FedSmall | FedClass |
|---|---|---|---|
| CIFAR-10 | **83** | 81 | 78.9 |
| CIFAR-100 | **48.5** | 42.5 | 37.6 |

tion of the fraction of the model that can be accommodated by the devices for both datasets. Our results demonstrate that FedSmall represents a robust benchmark for devices with lower capabilities. However, it imposes significant limitations on devices with higher capabilities by requiring them to train smaller models, even though they could potentially handle larger models. This can be problematic, particularly when there is a considerable disparity in capabilities between devices. In contrast, Federated FlexTrain does not penalize devices with higher capabilities, enabling them to maximize their potential and achieve superior results.

On the other hand, FedClass faces significant performance bottlenecks due to the data sharing issues. Since models are trained independently, and data is kept on the device's side, devices with higher capabilities cannot benefit from the data available on weaker devices. This can be especially challenging in scenarios with a large number of devices and non-i.i.d. data distribution.

However, FlexTrain circumvents this problem by enabling information sharing across different model configurations. By training a common model, the data on weaker devices can be used to train the initial layers of the global model, thereby improving the performance of the larger models. Furthermore, the proposed distillation technique enables smaller models to harness the advanced capabilities of more powerful devices, expanding their potential beyond their intrinsic limitations.

It is evident from our experiments that training multiple models in the centralized case resulted in a high accuracy but was limited by training time and energy consumption. However, in the federated setting, this approach quickly became disadvantageous in terms of accuracy due to the limitations of data sharing between devices with varying capabilities. These findings emphasize the need for a more flexible training approach that can consider these intricacies, a capability that Federated FlexTrain has demonstrated in our experiments. Our findings indicate that FlexTrain is a versatile approach that can be effectively applied in federated learning scenarios, offering several advantages. It incorporates weaker devices in the training process of the global model, which increases the amount of knowledge transferred to it and enhances the accuracy of the global model. Furthermore, it provides models to devices that meet their memory, energy, and latency constraints during the inference phase.

### 4.1 Experiments Details

To ensure the reproducibility of our experiments, we present a summary of the key details below. All experiments were conducted on Tesla-V100 GPUs, and the reported results are the averages obtained from running the experiments with three different seeds.

**Centralized Settings:** We used a fixed number of epochs, setting it to 160 for ResNets and 300 for the visual transformer. In each epoch, we performed a mini-batch stochastic gradient descent step with a batch size of 64. To apply regularization, we employed the stochastic depth procedure with a dropout probability of 0.5, weight decay of 0.001, momentum of 0.9, and learning rate of 0.05 for ResNet. For the visual transformer, we used dropout with probability 0.1 and a learning rate of 0.003. The distillation parameter, $\beta$, was set to 0.2 in both cases.

**Federated Settings:** In the federated settings, we conducted a total of 160 global communication rounds. Each communication round consisted of 10 local epochs. The settings for each local epoch were the same as those used in the centralized settings.

## 5 Discussions

**Summary.** We presented FlexTrain, a dynamic training framework designed to tackle the challenges posed by large machine learning models in heterogeneous devices environments. FlexTrain was shown to offer a good balance between accuracy and energy consumption in centralized settings, eliminating the need to train multiple sizes of models while maintaining high accuracy performance. Similarly, we demonstrated that the federated version of FlexTrain outperforms in terms of accuracy the approach of letting each class of devices, based on their capabilities, train a common model.

**Limitations.** FlexTrain adopts a deep-to-shallow deactivation strategy that facilitates the learning of basic features during the training process and allows deeper layers to build upon them and learn more complex features. This approach is particularly advantageous for residual neural architectures such as ResNets and Transformers. While these architectures are currently the state-of-the-art and widely used in various machine learning tasks, there may be some applications where different architectures are more suitable, which could limit the applicability of FlexTrain in those cases.

**Outlook.** As machine learning models continue to increase in size, the challenges related to them will become more pronounced. Given the widespread adoption of machine learning in various applications, it is crucial to address these challenges as weaker devices with limited computational and storage capabilities are likely to be involved. A promising approach to address these challenges is to combine various techniques proposed in the literature, including compression, quantization, sparsification, and more. Our FlexTrain approach provides an additional layer of flexibility that can be combined with these techniques to push the boundaries of efficiency in model training and deployment. We believe that this integrated approach will be essential to tackle the challenges related to large machine learning models and to support their deployment in heterogeneous devices environments.

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
