# OpenReview forum: "FlexTrain: A Dynamic Training Framework for Heterogeneous Devices Environments"
_NeurIPS.cc/2023/Workshop/WANT — WANT@NeurIPS 2023 Poster_

### Official Review · Reviewer_AoA9 · 2023-10-24
**A good quality paper highly relevant to this workshop.**

**Confidence:** 4

**Review:**

This 9-page article proposes a training framework for heterogeneous devices where the model trained locally depends on the computation and memory capabilities of the device while still being able to be used to improve the global model. This is done through a combination of active layer sampling and auto-distillation. The layer sampling is done so that basic features (first layers) are prioritized. Each model is trained using distillation on the next bigger model.
# Quality
The quality is good as the context is well presented, each technique is well explained, and the experiments are extensive and relevant. Energy consumption has been cited several times, but no measurements were done. I believe including an analysis of the energy consumed would significantly improve the quality of the paper, as it was proven that FLOPS are not correlated with energy consumption.
# Clarity
The paper is well organized. However, in my opinion, the experiment settings should be found before the description of the results.
# Originality
To my knowledge, both layer sampling and auto-distillation were not previously applied at the same time in both centralized and decentralized settings.
# Significance
The authors were able to significantly reduce the number of FLOPS (62% difference from single training and approximately 40% from independent training of various sizes) while achieving almost the same accuracy.
# Pros
- Highly reduced computations and training time
- Adaptative framework for heterogeneous devices
- experimentally validated with two different models on two datasets in both centralized and decentralized settings
# Cons
- Code is not open-sourced
- Restricted to resnet & transformers
- No analysis of the energy consumed.

I recommend to accept this paper.

---

### Official Review · Reviewer_hieG · 2023-10-25
**Interesting idea to obtain multiple sized models at once**

**Confidence:** 3

**Review:**

This paper proposes an interesting idea that can train multiple sized models at once. The main contribution is during training, one configuration is sampled for loss computation. The results also apply to federated learning which is another good use case of this technique. Overall I think this paper is a good fit for this workshop, but would like to raise the following two points:
1. how much extra training overhead is introduced by this approach?
2. It seems not quite necessary to sample configuration in the same manner in the paper. Usually configurations differing by 1 layer probably won't change the inference cost that much. Using sparser configuration sampling might speed up the training too.

---

### Official Review · Reviewer_2Bdc · 2023-10-26
**this paper proposes an very interesting DNN model training framework**

**Confidence:** 3

**Review:**

**Summary of the work**

Overall, I think this is a decent paper for WANT!

This paper proposes FlexTrain, a DNN model training framework that enables efficient deployment of deep learning models, while respecting device constraints, reducing communication costs, and ensuring smooth integration with diverse devices. The authors demonstrate the effectiveness of FlexTrain on the CIFAR-100 dataset. They also extend FlexTrain to the federated learning setting.

**Strength**

The studied problem is important, and the proposed algorithm is novel and efficient for training small and high performance DNN models. Specially, the paper proposes using active layers sampling and auto-distillation. The authors also extend FlexTrain to the federated learning setting, which is cable of sharing knowledge learned from the weaker devices to the more capable ones.

It is also great to see that with sacrificing only a small percentage of accuracy, a single global model trained with FlexTrain can be easily deployed on heterogeneous devices, which saves training time and energy consumption.

**Weakness**

The proposed algorithm is particularly advantageous for residual neural architectures such  as ResNets and Transformers. While these architectures. It is not clear if the algorithm works for some applications where different architectures are more suitable.

I would also like to see more theoretical analysis, especially in terms of the extension to federated learning.

---

### Meta-Review · Area_Chair_bpGx · 2023-10-27

**Recommendation:** Accept (Poster)
**Confidence:** 4

**Metareview:**

This paper proposes a dynamic training framework for heterogeneous devices with diverse storage and computational resources. All reviewers think this manuscript is highly relevant to the workshop and champion the acceptance.

---

### Decision · Program_Chairs · 2023-10-28

**Decision:**

Accept (Poster)

**Comment:**

We thank the authors for their time and contribution to WANT and we are pleased to share that after the reviewing process the paper has been accepted. Congratulations! We encourage the authors to consider reviewers' feedback for the improvement of the camera-ready version. We hope to see you in person at the workshop and brainstorm on efficient training research together!